# Sustainable polycarbonate adhesives for dry and aqueous conditions with thermoresponsive properties

Anjeza Beharaj [1], Ethan Z. McCaslin [1], William A. Blessing [1] & Mark W. Grinstaff [1*]

Pressure sensitive adhesives are ubiquitous in commodity products such as tapes, bandages, labels, packaging, and insulation. With single use plastics comprising almost half of yearly plastic production, it is essential that the design, synthesis, and decomposition products of future materials, including polymer adhesives, are within the context of a healthy ecosystem along with comparable or superior performance to conventional materials. Here we show a series of sustainable polymeric adhesives, with an eco-design, that perform in both dry and wet environments. The terpolymerization of propylene oxide, glycidyl butyrate, and $CO_2$, catalyzed by a cobalt salen complex bearing a quaternary ammonium salt, yields the poly (propylene-co-glycidyl butyrate carbonate)s (PPGBC)s. This polymeric adhesive system, composed of environmentally benign building blocks, implements carbon dioxide sequestration techniques, poses minimal environmental hazards, exhibits varied peel strengths from scotch tape to hot-melt wood-glue, and adheres to metal, glass, wood, and Teflon® surfaces.

[1] Departments of Chemistry, Biomedical Engineering, and Medicine, Boston University, Boston, MA 02215, USA. *email: mgrin@bu.edu

Pressure sensitive adhesives (PSAs), soft polymeric materials that adhere to surfaces via Van der Waals interactions under pressure, are primarily composed of acrylic copolymers and polystyrene/isoprene/butadiene based blends[1,2]. These viscoelastic polymers teeter the delicate balance of liquid enough to wet a surface and elastic enough to resist direction of motion. Viscoelastic fine-tuning of bulk polymer properties is accomplished through the addition of tackifiers[3,4], plasticizers[5,6], post-polymerization cross-linking[7,8], or the covalent combination of two or more distinct monomers[9–12].

Due to their ability to bond dissimilar materials without incompatibility concerns, PSAs are ubiquitous in commodity products such as tapes[13,14], bandages[15], labels[16], household decorations[17], and packaging[18]. Driven by high demand in consumer goods, the PSA market value is expected to reach $9.5 billion by 2024, with environmentally friendly PSAs representing the fastest growing technology segment[19]. Recent advances in this field of adhesion chemistry include biomimetic approaches[20] such as nanoscale fabrication of the fibrillous geometry found in the adhesive pads of gecko's feet[21–24], and utilization of dopamine enriched proteins as found in the adhesive footpad of marine mussels[25–30].

From a polymeric materials design perspective, we are cognizant of the need to consider the environmental impact of a polymer's lifecycle, the polymerization methodology, the composition of the building blocks, and the subsequent polymer breakdown products[31–33]. To this end, we utilize a synthetic pathway pioneered by Inoue et al.[34] and brought to realization by Coates[35] and Darensbourg[36], and a catalyst ligand framework optimized by Lu et al.[37,38], in which carbon dioxide and an oxiranyl monomer[39–42] are activated and linked together to afford a degradable polycarbonate.

Herein, we report a library of environmentally friendly, sustainable, strong, and responsive adhesives composed of carbonate terpolymers. These adhesives exhibit polymer compositional dependences on peel and tack strength, bind to metal, glass, wood, and polytetrafluoroethylene (PTFE), as well as exhibit reversible on-demand adhesion through a temperature trigger in both dry and wet environments. The synthetic approach is highly amenable to many oxiranyl monomers, including those derived from biological feed stocks lowering the dependency on petroleum, and allows for the fine-tuning of the polymer composition and microstructure to attain desired chemical, physical, degradation, and mechanical properties.

## Results

**Polycarbonate synthesis.** In order to mimic the pendant functionality of current commercial adhesives (Fig. 1a), we synthesized poly(propylene-co-glycidyl butyrate carbonate) (PPGBC) via the terpolymerization of glycidyl butyrate (GB), propylene oxide (PO), and 2.7 MPa of $CO_2$ catalyzed by a salen cobalt complex (2000:1 catalyst loading) at 40 °C (Fig. 1b, c, and Supplementary Methods). The ester side chain of GB imparts adhesivity through Van der Waals interactions, while PO allows for tighter compaction of polymer chains, raising the glass transition temperature, and polymeric cohesive strength. The monomeric units derived from chain scission of PPGBC are biologically benign[43,44] and are comprised of glycerol and PO, food additives identified as Generally Recognized as Safe (GRAS) by the FDA, as well as butyric acid, a compound responsible for the characteristic smell of feta cheese[45], and $CO_2$, an atmospheric gas (Fig. 1d).

Specifically, we synthesized a library of co- and terpolymers with varying monomeric feed ratios of GB and PO as shown in Table 1. The catalyst, $[S,S]$-[SalcyCo$^{III}$DNP]/DNP[38], polymerized PO with high turnover frequency (444 h$^{-1}$), high polymer

selectivity (>99%), moderate molecular weight (22 kg/mol), and low dispersity (1.18). Under the same conditions, the catalyst polymerized GB with lower TOF (77 h$^{-1}$), lower polymer selectivity (86%), lower molecular weight of (12 kg/mol), and similar dispersity (1.2). In the controlled and living $CO_2$/PO/GB terpolymerization, increasing the PO monomer feed concentration led to sequentially higher TOFs compared with GB alone. Similarly, increasing PO monomer feed concentration afforded greater molecular weight polymers and higher polymeric selectivity over the cyclic carbonate.

The glass transition temperature, measured by differential scanning calorimetry (DSC), is 28 °C and −7 °C for PO and GB, respectively. For the terpolymers, as the GB content increases the glass transition reduces from 0 to −30 °C. A bimodal distribution of chain length is observed for all polymers by GPC analysis, but dispersities remained low at ~1.2 (Supplementary Fig. 10). This observable phenomena is due to adventitious water molecules as MALDI-ToF spectroscopy revealed two initiating groups (hydroxyl and dinitrophenoxide) for polymeric chains and one terminating group (hydroxyl) (Supplementary Fig. 8).

Fineman–Ross analysis was undertaken to determine the probabilistic sequence distribution of monomers in the copolymer composition. In order to approximate steady state kinetics, the reactions were stopped at low conversions (~5%) and analyzed by $^1$H NMR (Supplementary Table 1). The Fineman–Ross linearization method revealed a strong correlation ($R^2 = 0.9994$) between monomer percentage in the feed and monomer incorporated into the polymer. The monomeric reactivity ratios for GB ($r_{GB} = k_{11}/k_{12}$) and PO ($r_{PO} = k_{22}/k_{21}$) are 1.32 and 0.26, respectively (Fig. 2), indicating consecutive incorporation of two GB units is more favored during the terpolymerization. This preference is due to the electron donating effects of the pendant ester of GB, increasing epoxide nucleophilicity over PO, and enabling faster coordination to the catalyst active site. Since the GB monomeric feed strongly resembles GB polymer incorporation at high conversions (Table 1, ~60% conversion), the terpolymer possesses a gradient distribution of PO insertion, with more PO units incorporated toward the end of the chain.

**Adhesion measurements.** We conducted peel testing to compare the relative adhesive strengths, defined as the force per width required to separate a flexible substrate from a rigid substrate, for all the viscous terpolymers at room temperature (Fig. 3a). Glass slides ($SiO_2$) were used as the base substrate, and A4 paper (2.6 × 8 cm), wetted with neat adhesive, was used as the face substrate. Testing was performed on an Instron 5944 series at 180° peel angle at a rate of 360 mm/min following procedures from ASTM D903.

PPGBC-56 exhibits superior adhesion with a peel force of 4.9 ± 0.41 N/cm compared with all other terpolymers and similar to Duct-tape® (3M 2929) (4.1 ± 0.48 N/cm). PGBC-100 (2.6 ± 0.23 N/cm) displays a peel strength comparable to Scotch-tape® (3M 810) (2.1 ± 0.20 N/cm). PPGBC-74 and PPGBC-87, exhibiting low glass transition temperatures (approximately −30 °C), both demonstrate comparably low peel strength of ~0.8 N/cm (Fig. 3a), about twice that of a Post-It® note (~0.4 N/cm) under the same experimental conditions[40]. All the polymer adhesives cohesively fail due to weaker bulk forces than surface bonding forces.

To assess PPGBC-56's adhesiveness to chemically distinct materials, we performed probe-tack testing on metal, glass, wood, and PTFE, using a DHR-2 rheometer at room temperature. The top 8 mm diameter steel plate (50.3 mm$^2$ surface area) was lowered at a rate of 100 μm/s unto one of the four adhesive coated substrates with an applied axial force of 50 N. After 5 s of contact,

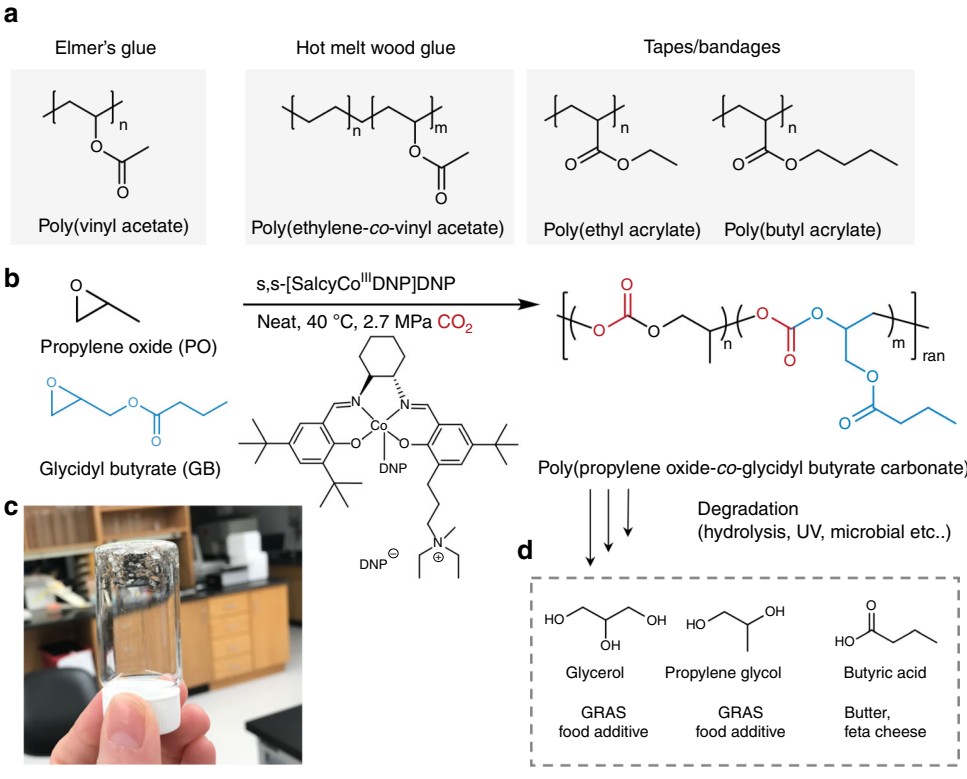

**Fig. 1** Chemical structures of commercial adhesives (**a**), and carbonate terpolymers (**b**). **c** Image of PPGBC, a clear highly viscous liquid. **d** Final products of complete polymer decomposition.

**Table 1 Terpolymerization of GB/PO/CO₂ catalyzed by (S,S)-[SalcyCo$^{III}$DNP]DNP.**

| Entry | $f_{GB:PO}$[a] | $F_{GB:PO}$[b] | TOF (h$^{-1}$)[c] | Selectivity (%)[d] | Tg °C[e] | Mn (kg/mol)[f] | Đ ($M_w/M_n$)[f] |
|---|---|---|---|---|---|---|---|
| PGBC-100 | 100:0 | 100:0 | 74 | 85 | −7 | 12.4 | 1.20 |
| PPGBC-87 | 90:10 | 87:13 | 78 | 86 | −30 | 6.1 | 1.12 |
| PPGBC-74 | 70:30 | 74:26 | 129 | 82 | −29 | 6.0 | 1.10 |
| PPGBC-56 | 50:50 | 56:44 | 144 | 95 | −9 | 8.6 | 1.08 |
| PPGBC-33 | 30:70 | 33:67 | 158 | 90 | −12 | 13.6 | 1.11 |
| PPGBC-22 | 10:90 | 22:78 | 225 | 99 | 0 | 9.6 | 1.11 |
| PPC-100 | 0:100 | 0:100 | 444 | >99 | 28 | 22 | 1.18 |

The reaction was performed in neat epoxide (10 mmol) in a 15 mL autoclave under 2.7 MPa CO₂ pressure at 40 °C with 2000:1 catalyst loading
[a]Molar ratio of monomer feed
[b]Molar ratio of monomer incorporation in polymer chain
[c]Turnover frequency (TOF) = mole of product (polycarbonates)/mol of cat. per hour. Reactions were stopped at ~60% conversion
[d]Percent of polymer formed vs. cyclic carbonate as determined by ¹H NMR
[e]Determined by DSC analysis
[f]Determined by gel permeation chromatography in THF, calibrated with polystyrene standards

the top steel substrate was pulled apart with a rate of 100 μm/s, and the tack strength (S$_{tack}$), defined as the peak of the force curve, was measured. PPGBC-56 possesses a similar tack strength of ~41 N to metal, wood, and steel (Fig. 3b). A reduced but still substantial tack strength is observed for PTFE of 27 ± 1.8 N. PPGBC-56 cohesively fails to metal, wood, and steel while adhesively fails to PTFE. The lower adhesive strength is likely due to the weaker Van der Waals forces between the adhesive and PTFE from the high electronegativity of the fluorine atoms.

In order to identify a temperature responsive PSA, we assessed the adhesion energy of all the polymer formulations at 37 and 50 °C, using a metal-on-metal probe-tack testing protocol. Of the formulations, PPGBC-56 exhibits the desired tack profile for bonding at 37 °C and debonding at 50 °C (Supplementary Figs. 12 and 13). Specifically, as shown in Fig. 3c, the tack strength of PPGBC-56 with 1 N of applied axial pressure and 5 s dwell time

at 20 °C is 9.0 ± 1.8 N in a dry environment, and the PSA exhibits adhesive failure with debonding occurring at the polymer–metal interface. Raising the temperature to 37 °C significantly increases the S$_{tack}$ to 30.4 ± 5.2 N. At the higher temperature, the polymer flows, spreads across, and wets the metal increasing the surface area of interaction and adherence to the metal rod surface. The interfacial adhesion between the two surfaces increases such that the PSA now fails cohesively (weaker interchain Van der Waals forces), and debonding occurs between the polymer strands. A further increase in temperature to 50 °C reduces the S$_{tack}$ to 9.6 ± 1.6 N and by 100 °C, the tack strength is significantly less and 3.3 ± 0.2 N. As the temperature rises, interchain Van der Waals forces further weaken due to increased volume and mobility between polymer strands, and the energy required to debond the materials decreases. The same trend is observed with the PPGBC-56 in an aqueous environment with diminished tack strength overall (Fig . 3c).

A material's viscosity directly correlates with its timely ability to wet a surface and subsequently form an adhesive bond. To quantify the relationship between the pressure applied to PPGBC-56 and its ability to form a strong adhesive bond, we conducted metal-on-metal tack testing with varying applied axial forces at 20 and 37 °C using an 8 mm steel parallel plate geometry. Again, the probe's dwell time was 5 s and the top plate was pulled apart with a rate of 100 μm/s.

At a temperature of 20 °C, a strong correlation between applied axial force and peak tack strength is observed for PPGBC-56

(Fig. 3d). At a low applied force, 0.5 N, the $S_{tack}$ is $1.49 \pm 0.53$ N and the material exhibits adhesive failure. As the normal force applied increases from 1 to 5 to 20 N, the $S_{tack}$ then increases from $9.03 \pm 1.83$ to $17.5 \pm 2.51$ to $25.0 \pm 1.31$ N. At 50 N applied axial force, the debonding force is greater than the maximum load cell of the rheometer (55 N), and thus, the $S_{tack}$ is estimated to be >55 N. At a temperature of 37 °C, a significantly different tack profile is observed. Even 0.5 N of applied axial force is sufficient to achieve maximum tack strength. As the polymer's viscosity is significantly less at this temperature, with a stronger viscous than

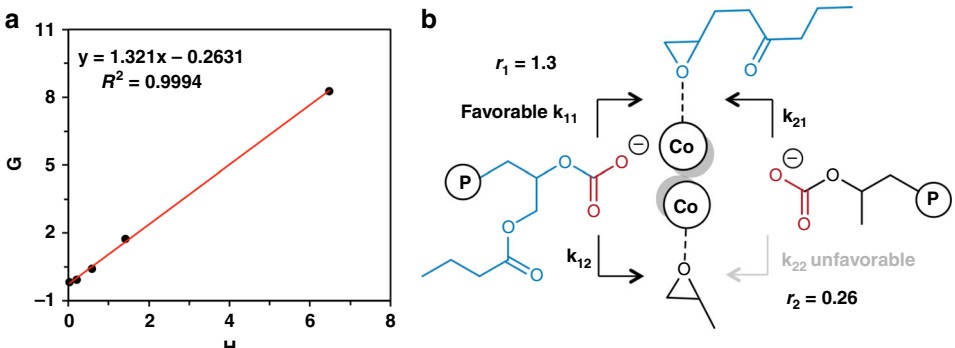

**Fig. 2** Probabilistic sequence distribution of monomers in the terpolymer. **a** Fineman–Ross plot of PPGBC at low conversions. $X = [f_{[GB]}/f_{[PO]}]$, $Y = [F_{[GB]}/F_{[PO]}]$, $H = Y^2/X$, $G = Y(X-1)/X$. **b** Schematic of propagation reactions. Source data are provided as a Source Data file and data for Fig. 2a is provided in Supplementary Table 1.

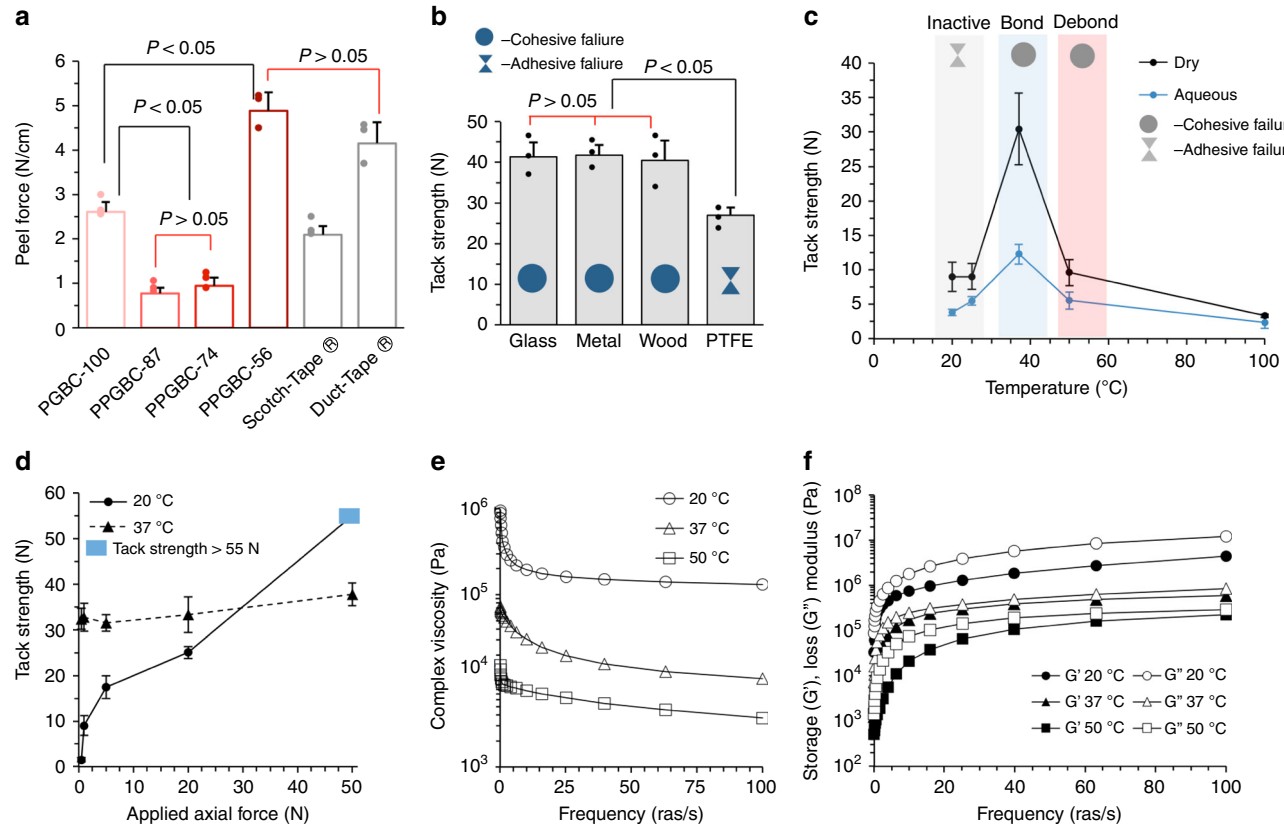

**Fig. 3** Adhesive and rheometric properties of terpolymers under various conditions. **a** Peel testing (180°) at room temperature (22 °C) of viscous poly (propylene-co-glycidyl butyrate carbonate)s and commercial adhesives ($n = 3$). **b** Tack strength of PPGBC-56 applied to four surfaces at room temperature ($n = 3$). **c** Tack strength of PPGBC-56 with 1 Newton of applied axial force at different temperatures tested in atmospheric conditions and underwater ($n = 3$). **d** Tack Strength vs. different applied axial pressure for PPGBC-56. **e** Frequency sweep of the complex viscosity (η) of PPGBC-56 at three different temperature ranges ($n = 3$). **f** Frequency sweep of the storage (G′) and loss (G″) modulus of PPGBC-56 at 20, 37, and 50 °C. Source data are provided as a Source Data file for Figs. 3-d. Error bars indicate mean ± s.e.m. and all significance testing was conducted using ANOVA.

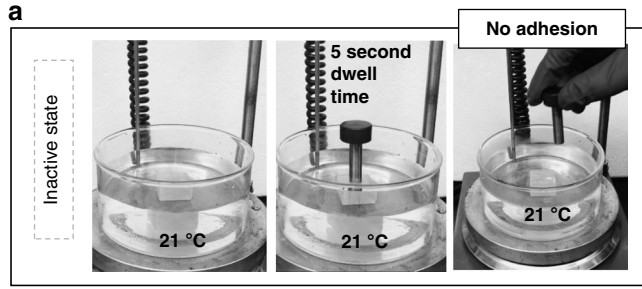

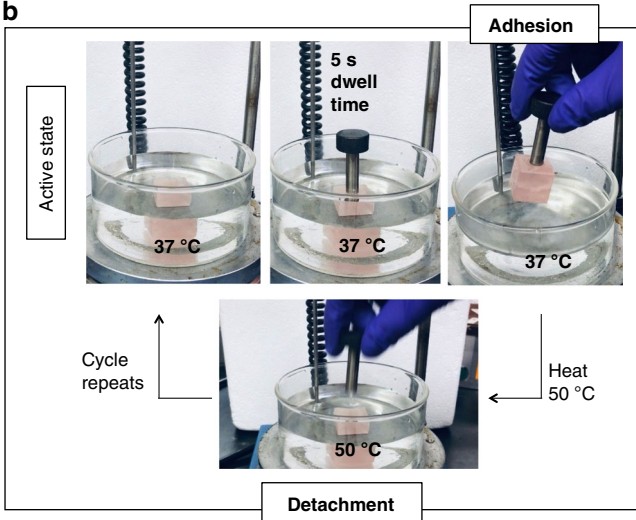

**Fig. 4** Adhesive system with thermoresponsive bonding and debonding. **a** Glass cube (20 g) coated with PPGBC-56 in DI water with a steel rod (35 g, 50.3 mm² surface area) placed on top. At 21 °C, the adhesive is not able to bond to the rod and the adhesive system is inactive. **b** At 37 °C, there is sufficient bonding to pick up the glass cube. At 50 °C, the adhesive weakens and detachment of the cube from the rod occurs. Lowering the temperature to 37 °C repeats the bonding/debonding cycle. Lowering the temperature further to 21 °C returns the system to the inactive state.

elastic profile (Fig. 3e, f), PPGBC-56 wets and strongly adheres to the probe's surface with minimal applied force (i.e., essentially independent of applied force). An applied axial force of 0.5 N requires 32.1 ± 2.3 N of force to separate the materials. Increasing the applied force to 1, 5, and 20 N does not change the peak debonding force, which remains ~31 N. At 50 N of applied axial force, the $S_{tack}$ force increases to 37.0 ± 2.51 N, although bearing no statistical significance.

Utilizing this information we hypothesize that a local temperature change will trigger adhesion by controlling the viscoelastic state of the adhesive underwater. One side of a 1 in.³ glass cube (SiO₂, 20 g) was coated with PPGBC-56 and immersed in 21 °C DI water. A 35 g metal rod with a surface area of 50.3 mm² (8 mm diameter) was gently placed on the adhesive surface, let stand for 5 s, and subsequently removed as shown in Fig. 4a. Under these conditions, the metal rod does not adhere to the glass cube (Supplementary Movie 1). Upon heating the water to 37 °C, the same metal rod was gently placed on the adhesive coated glass cube surface, let stand for 5 s, and removed from the water as shown in Fig. 4b. At this temperature, the metal rod binds to the glass with sufficient force that it is able to pick up and hold the glass cube (Supplementary Movie 2). Raising the temperature of the water to 50 °C detaches the metal rod from the glass cube (Supplementary Movie 3). At this higher temperature, heating the system expands the polymeric volume, and releases the rod as intermolecular Van der Waals forces wane and

cohesive failure detaches the cube. Repeated 37 and 21 °C cycles of the rod/cube system display reversibility with the same adhesive coating attaching and detaching the rod. The adhesive detaches through cohesive failure, but remains on each respective surface and polymeric mass is not lost into the water. Replacing PPGBC-56 with duct tape at either 21 or 37 °C did not result in bonding between the metal rod and glass cube, likely because the applied axial force (weight of the rod) is not sufficient to induce spreading and contact bonding of the adhesive.

**Cytotoxicity studies.** Although these polycarbonates are composed of relatively benign building blocks, evaluation of toxicity is warranted with the synthesis of new materials, especially, given their potential use in the consumer goods (e.g., food packaging) and medical device areas (e.g., pressure sensitive adhesive). Preliminary in vitro transwell cytotoxicity studies with PPGBC-56 and NIH 3T3 fibroblasts demonstrate that after 24 h, minimal cytotoxicity is observed even at concentrations as high as 20 mg/ mL of terpolymer (>87% viability; Supplementary Fig. 15). In addition, the reported $LD_{50}$s of the degradation products butyric acid and glycerol are 3.7–9.8 g/kg (European Chemicals Agency) and 4.42 g/kg in rabbits (National Library of Medicine: TOX-NET), respectively. Similarly, exposure of RAW 264.7 macrophages to PPGBC-56 for 24 h affords an absence of the proinflammatory cytokine IL-6 up to concentrations of 20 mg/ mL, as the expressed cytokine IL-6 levels are comparable to the negative control (Supplementary Fig. 16).

In conclusion, we synthesized a series of polycarbonate terpolymers using a cobalt(III) salen catalyst in high turnover frequency, high polymer selectivity, moderate molecular weight, and low dispersity. Being composed of building blocks known to be on the GRAS list, present in foods, or our atmosphere, these terpolymers are attractive materials for potential commercial use from both environmental and biomedical perspectives. Of the synthesized polymers, PPGBC-56 exhibits stronger adhesion than commercial Scotch-tape® and comparable adhesion to Duct-tape®. Furthermore, this adhesive sticks to a variety of chemically distinct materials. At 20 °C, increased applied pressure yields greater tack strength force. This dependency is absent at a high temperature of 37 °C, and the adhesive itself is able to wet and subsequently bond surfaces with minimal applied contact force and time. The high viscosity of the adhesive at room temperature enables a thermoresponsive temperature trigger of adhesion to induce bonding and debonding. Through judicious choice of polymer with an eco design, our approach will open new avenues of research as well as catalyze the investigation of unique functional adhesive materials to meet the ever-increasing demands from society.

## Methods
**General information.** All manipulations involving air- and/or water-sensitive compounds were carried out in a glovebox. All oxiranyl monomers were refluxed over CaH₂, and fractionally distilled under a nitrogen atmosphere prior to use. Carbon dioxide (99.995%, bone dry) was purchased from Airgas and used as received. Reagents were purchased from Sigma-Aldrich and used as received. All measurements were taken from distinct samples.

**Materials characterization.** The characterization and synthesis of all compounds are described in full detail in the Supplementary Information (Supplementary Figs. 1–7). Materials previously synthesized and characterized are provided with references to the original work.

**NMR experiments.** ¹H and ¹³C NMR spectra were recorded on a Varian 500 MHz type (¹H, 500 MHz; ¹³C, 125 MHz) spectrometer. Their peak frequencies were referenced against the solvent, chloroform-d at δ 7.24 for ¹H NMR and δ 77.23 ppm for ¹³C NMR, respectively.

**Gel permeation chromatography**. All polymer molecular weights were determined by gel permeation chromatography versus polystyrene standards (Agilent Technologies) using THF as the eluent at a flow rate of 1.0 mL/min through a Styragel column (HR4E THF, 7.8 × 300 mm) with a Waters 2414 refractive index detector.

**MALDI-ToF**. MALDI-ToF mass values for polymers were determined using a Bruker autoflex Speed SMART MALDI-ToF mass spectrometer equipped with a SMART-beam II and a flash detector. Samples were prepared by dissolving in a 1:1 vol/vol mixture of matrix solvent (10 mg/mL solution of dithranol in THF with 0.1% AgTFA) and 10 mg polymer dissolved in minimal amount THF.

**DSC**. The thermal properties of the polymers were measured by DSC using a TA Q100 under a nitrogen atmosphere (nitrogen flow rate: 60 mL min$^{-1}$). All samples were tested at a heating rate of 10 °C/min and a cooling rate of 10 °C/min from −40 to 80 °C. The weight of all samples was between 2 and 10 mg and the samples underwent three heat-cool-heat cycles. The glass transition temperature, Tg, was noted in the DSC-thermogram as the midpoint temperature of the glass transition peak in the second heating cycle (Supplementary Fig. 9).

**180° peel strength**. The peel adhesion test was carried out at room temperature (22 °C) by using *Fischerbrand* glass microscope (SiO$_2$) slides (base stock) and a A4 paper (face stock) as substrates. (Supplementary Fig. 11).

The face dimensions for the glass slides were 7.6 × 2.6 cm. The adhesive was coated on the nonfrosted surface of the glass plate containing a coating area of 2 × 2.6 cm with a coating thickness of ~30 μm. Then, the paper substrate was stuck on the coated glass slide with moderate human finger pressure. The sample was let to settle for 1 min prior to testing on an Intron 5944 with peel speed operating at 360 mm/min.

Commercial all-purpose Duct Tape® (3M 2929) and Scotch Tape® (3M 810) were used as received, (besides width modifications) and stuck to the glass. Duct tape was cut to 2.6 cm of width, scotch tape was not modified. Three separate specimens were used for each adhesive formulation in this test. The average peak from the load propagation graph was used to calculate the peeling force. Peel strength is defined as the average load per width of the bondline required to separate progressively a flexible member from a rigid member (ASTM D 903).

**Probe tack**. All tack testing were performed on a Discovery Hybrid Rheometer (DHR-2 series) with 8 mm stainless steel-sand blasted parallel plate geometry with a Peltier plate. The adhesive was placed on the bottom plate and a top probe moving at 100 μm/s rested on the adhesive until the desired axial force was reached. After 5 s, the top probe pulled away at a rate of 100 μm/s (Supplementary Fig. 12a and b). The peak of the force curve is defined as the tack strength ($T_s$) and the area under the curve is defined as the tack energy ($T_e$) as calculated by Eq. (1), where $A$ denotes the surface area (m$^2$) of the probe, $r$ is the rate of probe separation in debonding (m/s), $F$ is the force (N) measured during debonding, and $t$ is the time in seconds;

$$T_e = 2x\left[\frac{r}{A}\int_{ti}^{tf} F(t)dt\right] \tag{1}$$

**Frequency sweeps**. All oscillatory sweeps were performed on a Discovery Hybrid Rheometer (DHR-2 series) with 8 mm stainless steel parallel plate geometry with a gap size of 50 μm. Frequency sweeps were performed from 0.1 to 100 rad/s or 1 to 500 rad/s at 1% strain (determined to be in the linear viscoelastic region with a previous strain sweep) at specified temperatures (20, 25, 37, and 50 °C) controlled by a Peltier plate. (Supplementary Fig. 14).

**Cell culture**. NIH 3T3 murine fibroblasts (ATCC) were cultured in Dulbecco's modified Eagle's medium supplemented with 10% bovine calf serum and 1% penicillin–streptomycin. RAW 264.7 murine macrophages were cultured in Dulbecco's modified Eagle's medium supplemented with 10% fetal bovine serum and 1% penicillin–streptomycin. Cells were maintained in a sterile, humidified environment at 37 °C with 5% CO$_2$.

**In vitro evaluation of cytotoxicity**. NIH 3T3 cells were seeded in a 96-well plate at a density of 20,000 cells/well and were allowed to adhere for 24 h. The media was then replaced with fresh media, and cells were incubated with polymer samples in 5% DMSO using transwell inserts (0.4 μm pores). Cell viability was assessed 24 h after treatment via the MTS in vitro cytotoxicity assay (CellTiter 96 Aqueous One, Promega). The average of two experiments ($n = 2$) in which $n = 6$ per polymer concentration.

**In vitro evaluation of immunogenicity**. RAW 264.7 cells were seeded in a 96-well plate at a density of 30,000 cells/well and were allowed to adhere for 24 h. The media was then replaced with fresh media, and cells were incubated with polymer samples in 5% DMSO using transwell inserts (0.4 μm pores). IL-6 levels were measured via ELISA kit (Abcam) and compared with those of RAW 264.7 treated with lipopolysaccharide—a molecule known to stimulate IL-6 production and immunogenicity in vitro ($n = 6$).

**Reporting summary**. Further information on research design is available in the Nature Research Reporting Summary linked to this article.

## Data availability
The data supporting this article are found within the text and the supplementary information file. The source data underlying Figs. 2a, 3a–d, and Supplementary Fig. 12 are provided as a Source Data file (https://doi.org/10.6084/m9.figshare.10013315). Any additional data is available from the corresponding author upon request.

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

## Acknowledgements

This work was supported in part by the NSF (DMR-1410450 and DMR-1507081) and the BU Undergraduate Research Opportunities Program (UROP; EZM). NMR facilities at Boston University are supported by the NSF (CHE-0619339).

## Author contributions

A.B. contributed to the synthesis of polymers, the adhesion measurements, and the analysis of data. E.Z.M. contributed to synthesis and characterization of the polymers. W. A.B. contributed to the in vitro experiments. A.B. and M.W.G. contributed to the experimental design and wrote the paper.

## Competing interests

The authors declare no competing interests.
