## [Transparent Peer Review File · Nature Communications]

Reviewers' comments:

Reviewer #1 (Remarks to the Author):

I have read the manuscript entitled "Sustainable Polycarbonate Adhesives for Dry and Aqueous Conditions with Thermo-responsive Properties" from Prof Grinstaff and colleagues.

Authors describe the synthesis of a polymer library of polymer adhesives that perform in both dry and wet environments.

These compounds represent indeed interesting biodegradable polymers with natural degradation products very well illustrated in Fig 1.

Experimental work is well designed and performed including an intensive chemical and physico-chemical characterization. The manuscript is clearly written, the figures and their legends are clear and the main conclusions are correctly drawn from the observations. Movies are useful elements in the demonstration of polymers performance.

Nevertheless, there are few points that need to be addressed as follows:

1- As shown in Figure 1a only two adhesives from the library were associated with significant peel force. It would be of interest to add a "non-adhesive" control in order to conclude about the observation of low adhesives PPGPC-87 and PPGPC-74.

2- Page 4-lines 89-90: Authors state "A bimodal distribution of chain length is observed for all polymers by GPC analysis, but disparities remained low at ~1.2 (Figure S8)".

a. How it can be possible to obtain such low poly dispersity index (PDI) with bimodal distribution (two population polymer chain lengths).

b. The argument that authors give using MALDI-ToF data explains indeed the bimodal distribution observed by GPC but this still doesn't explain the low value of PDI (~1.2).

3- Page 4 Lines 92-93: Authors state: MALDI-ToF spectroscopy revealed two initiating groups (hydroxyl and dinitrophenoxide) for polymeric chains FigS6".

What is the origin of the dinitrophenoxide group? It's not present neither in the reagents nor in the structure of the ligand cobalt catalyst.

4- In vitro biological testing: As preliminary investigation the choice of cytotoxicity and IL-6 expression level is reasonable, but :

a. I'm surprised by the density of cells used in both tests (NIH 3T3 cells: 96-well plate at a density 20,000 cells/well and RAW 264.7 cells: a 96-well plate at a density of 30,000 cells/well). The normal seeding in 96 well plate is 5000 – 10000 Maximum).

b. The polymer that was tested and whose result is reported in Figures S13 and S14 is not specified out of the library reported in the manuscript, neither in the main text nor in figures ligands.

c. The detection of IL-6 is a good choice but we can't speak about the "absence immunogenicity" with the only result of IL-6 levels. This "preliminary result" may indicate the absence of high local inflammation reaction but we can't generalize the conclusion into "no immunogenicity" especially that the IL-6 expression was defined at one time point chosen arbitrary (24h).

d. It's confusing that the cell viability is around 60% (with as much low Standard deviation) at 0.1 mg/mL and then goes up into around 80 % at 10 and 20 mg/mL !!! . Could this result be related to the excessive number of cells in the wells that disturb the test realization. I suspect the formation of various cell layers with such high intensity that may disturb the biochemical MTS test.

e. It will be of interest to define the context of these biological evaluation: is it for potential use in medical devices? In biomaterials conception? for food packaging ...etc.

Minor points

1- Page 3 Line 62 : "...including those derived from biological feed stocks "lessening" the dependency"

Please replace the term "lessening" by lowering

2- SI Page 1 line 14-15: "Reagents were purchased from Sigma Aldrich and used as received. Propylene Oxide was purchased from Sigma Aldrich".

No need to mention Sigma Aldrich twice

3- SI In the ligand of Figure S7 please replace the term: "DSC trace" by "DSC thermogram"

4- SI Page 2 DSC paragraph: Authors state that "The measurements were performed from $-70\text{ }^{\circ}\text{C}$ to $125\text{ }^{\circ}\text{C}$ at heating and cooling rates $-70\text{ }^{\circ}\text{C}$ to $125\text{ }^{\circ}\text{C}$ and 2 lines further they state that "All samples... were tested from -60 to $100\text{ }^{\circ}\text{C}$ ". In figure S7 (DSC trace) the thermograph shown is from -50 to $80\text{ }^{\circ}\text{C}$.

These values need to be homogenized. This paragraph needs to be properly reedited. There is multiple repetitions.

5- SI: In all NMR figures please add the term spectrum (NMR spectrum).

Reviewer #2 (Remarks to the Author):

This manuscript by Grinstaff and coworkers describes an interesting and important study regarding the thermo-responsive sustainable polycarbonate adhesives, prepared from the terpolymerization of propylene oxide, glycidyl butyrate, and CO_2 using a bifunctional cobalt catalyst. Of the synthesized terpolymers, PPGBC-56 exhibited stronger adhesion than commercial scotch tape and comparable adhesion to duct tape. Notably, this adhesive sticks to a variety of chemically distinct materials, and its high viscosity at ambient temperature enables a thermo-responsive temperature trigger of adhesion to induce bonding and debonding. The work was well done and good described. Therefore, I would like to recommend its publication in Nature Communication as it is or after minor revision.

Minor points:

1. From the results in Table 1 at different monomer feed ratios of propylene oxide vs. glycidyl butyrate, glycidyl butyrate exhibited higher reactivity than propylene oxide during their terpolymerization with CO_2 using the bifunctional cobalt catalyst. On the contrary, propylene oxide showed significantly greater reactivity than glycidyl butyrate in the copolymerization with CO_2 . It means that the presence of propylene oxide enhances the reactivity of glycidyl butyrate, while glycidyl butyrate significantly inhibits the reactivity of propylene oxide. The authors should give a possible explanation.
2. For a comparison purpose, it is suggested to synthesize PPC-b-PGBC block terpolymer for testing its adhesive property.
3. Did the authors attempt other epoxides for the substitution for propylene oxide in the terpolymerization?
4. The used bifunctional cobalt catalyst in this study was first reported in 2009 (J. Am. Chem. Soc. 2009, 131, 11509–11518). Therefore, the supplemental citation of the literature is suggested.

Reviewer #3 (Remarks to the Author):

The manuscript of Prof. Grinstaff et al. reports on the library of new, environmentally friendly, sustainable, strong, and responsive adhesives composed of carbonate terpolymers. The reliable bonding between soft tissues is challenging because of the wet and complex environment and geometries. Also available tissue adhesives do not meet basic demands strong adhesion and cohesion; controlled and precise delivery; biocompatibility and biodegradability. Therefore, the topic of the work on development of poly(propylene-co-glycidyl butyrate carbonate)s as sustainable polymer adhesives with an eco-design and performance in both dry and wet environments is very timely and crucially important. Important is that this polymer is non-toxic and non-immunogenic in vitro. Also the aspects of adherence to a variety of substrates such as metal, glass, wood, and Teflon® surfaces are imperative. Furthermore, reversibility and T-responsiveness of bonding between $21\text{ }^{\circ}\text{C}$ and $37\text{ }^{\circ}\text{C}$ are addressed. The authors also have shown that adhesion increased at $37\text{ }^{\circ}\text{C}$ and is lost at $50\text{ }^{\circ}\text{C}$. This effect should be more clearly explained in the discussion and conclusion part before publishing.

Point by Point Response to Referees' Comments

Reviewer #1 (Remarks to the Author):

I have read the manuscript entitled "Sustainable Polycarbonate Adhesives for Dry and Aqueous Conditions with Thermo-responsive Properties" from Prof Grinstaff and colleagues.

Authors describe the synthesis of a polymer library of polymer adhesives that perform in both dry and wet environments.

These compounds represent indeed interesting biodegradable polymers with natural degradation products very well illustrated in Fig 1.

Experimental work is well designed and performed including an intensive chemical and physico-chemical characterization. The manuscript is clearly written, the figures and their legends are clear and the main conclusions are correctly drawn from the observations. Movies are useful elements in the demonstration of polymers performance.

Nevertheless, there are few points that need to be addressed as follows:

Response: We thank the referee for their positive comments on the importance of our work and the clarity of text and presentation of the findings.

Comment 1- As shown in Figure 1a only two adhesives from the library were associated with significant peel force. It would be of interest to add a "non-adhesive" control in order to conclude about the observation of low adhesives PPGBC-87 and PPGBC-74.

Response 1: We thank the referee for his/her question. We believe they mean figure 3a, describing the peel testing conducted on the polymers. In these experiments, PPGBC-100 possesses comparable peel strength to scotch tape, and PPGBC-56 possesses comparable peel strength to duct tape. PPGBC-87 and PPGBC-74 both exhibit peel strength around 1 N. For a comparison to a weak adhesive or "non-adhesive", the peel strength of a post-it note under the same experimental parameters is ~0.4 N. Thus, PPGBC-87 and 74 possess peel strength comparable to that of twice a post-it note. Without an adhesive present the peel strength is zero.

We have amended page 5 lines 118-119 to state "PPGBC-74 and PPGBC-87, exhibiting low glass transition temperatures (~ -30 °C), both demonstrate comparably low peel strength of ~0.8 N/cm (**Figure 3a**), about twice that of a Post-It[®] note (~0.4 N/cm) under the same experimental conditions."⁴⁰ (New Citation included).

Comment 2- Page 4-lines 89-90: Authors state "A bimodal distribution of chain length is observed for all polymers by GPC analysis, but disparities remained low at ~1.2 (Figure S8)".

a. How it can be possible to obtain such low poly dispersity index (PDI) with bimodal distribution (two population polymer chain lengths).

b. The argument that authors give using MALDI-T of data explains indeed the bimodal distribution observed by GPC but this still doesn't explain the low value of PDI (~1.2).

Response 2: This is an important question, and we thank the referee for allowing us to clarify the mechanism of polymerization for our novel materials. The PDI is low because this is a controlled and living polymerization (PDI range 1.01-1.3) and initiation is faster than propagation.

Once all the initiating groups (DNP) have started chains for polymerization, the limiting factor in the increase of MW is the catalyst's turn over frequency. Each chain must associate with a catalyst moiety to increase in length by one monomer. This process would lead to a monomodal distribution with a small PDI as there is no chain favorability for the catalyst (same chemical environment for all end groups) and chain increase will follow a statistical distribution.

However, in the case where water molecules (or any alcohol) are present, a hydrogen atom from water will protonate the active anionic carbonate propagating end-group, temporarily

terminating the polymer chain. This leads to a newly formed hydroxyl initiating group attacking a monomeric epoxide and starting a new polymer chain.

As these species begin to propagate and increase chain length, the hydroxyl initiating polymer can associate with the catalyst from both ends (as both are hydroxyl species), while the DNP can only propagate from one end (the terminal hydroxyl). This leads to the polymer with the hydroxyl initiator to have higher molecular weight and a bimodal distribution appears.

However, since both MWs are fairly controlled and overlapping, the PDI still remains low. We report the PDI as the calculation spanning the two peaks, which result in PDI ranges from 1.1-1.2. Calculating over each peak on its own results in PDI ranges of 1.04-1.05.

For more information on this phenomenon, Darensbourg published a noteworthy paper detailing the process further. (J. Chem. Educ. 2017, 94, 1691-1695). Additionally, we have stated this is a controlled and living polymerization. Page 3, line 85.

Comment 3- Page 4 Lines 92-93: Authors state: MALDI-ToF spectroscopy revealed two initiating groups (hydroxyl and dinitrophenoxide) for polymeric chains FigS6”.

What is the origin of the dinitrophenoxide group? It's not present neither in the reagents nor in the structure of the ligand cobalt catalyst.

Response 3: Thank you for pointing out this oversight. The dinitrophenoxide is symbolized by the acronym DNP. It's the axial ligand in the cobalt catalyst which initiates polymerization. We have edited the SI page 5 section 2.2 to include the full name and structure of the molecule for clarification.

Comment 4- In vitro biological testing: As preliminary investigation the choice of cytotoxicity and IL-6 expression level is reasonable, but:

a. I'm surprised by the density of cells used in both tests (NIH 3T3 cells: 96-well plate at a density 20,000 cells/well and RAW 264.7 cells: a 96-well plate at a density of 30,000 cells/well). The normal seeding in 96 well plate is 5000 – 10000 Maximum).

Response 4a: This seeding density is on the high end due to the 24-hour period of the experiment. We did not anticipate the cells to divide substantially, and we wanted to observe a strong signal-to-noise ratio. Using a seeding density of 5,000 cells per well did not provide this for us in our experimental time frame. With 20,000-30,000 cells per well, there was no observed cell death or cells “floating” in the wells prior to the experiment.

b. The polymer that was tested and whose result is reported in Figures S13 and S14 is not specified out of the library reported in the manuscript, neither in the main text nor in figures ligands.

Response 4b: We thank the referee for pointing out this omission. The polymer tested is PPGBC-56. We have changed the title of SI Figures 13 and 14 to include that information.

c. The detection of IL-6 is a good choice but we can't speak about the “absence immunogenicity” with the only result of IL-6 levels. This “preliminary result” may indicate the absence of high local inflammation reaction but we can't generalize the conclusion into “no immunogenicity” especially that the IL-6 expression was defined at one time point chosen arbitrary (24h).

Response 4c: We agree with the referee that “absence of Immunogenicity” is indeed an overreach of the experiment as any *in vitro* experiment cannot sufficiently determine the immunological response of a reagent. Determining secreted IL-6 levels is a common and facile method of an *in vitro* test for immunogenicity without using an animal model. We have altered the statement on page 8 line 195-197 to state “absence of the pro-inflammatory cytokine IL-6”.

d. It's confusing that the cell viability is around 60% (with as much low Standard deviation) at

0.1 mg/mL and then goes up into around 80 % at 10 and 20 mg/mL !!! . Could this result be related to the excessive number of cells in the wells that disturb the test realization. I suspect the formation of various cell layers with such high intensity that may disturb the biochemical MTS test.

Response 4d: We have repeated the experiment and combined the data sets (N=2, and n=6 per concentration; thus n=12 averaged). The new plot is included in the SI and the variability in cell viability is less across the concentration range. The reviewer maybe right that the cell layers may disturb the biochemical MTS test and contribute to the standard deviation error bars. None the less, minimal cytotoxicity is observed (>87% viability) at all polymer concentrations and even as high as 20 mg/mL. We have replaced “no cytotoxicity” with “minimal cytotoxicity” on page 8, line 193. Additionally, we added text to include the reported LD50s of the degradation products butyric acid (3.7 – 9.8 g/kg, rabbit, European Chemicals Agency) and glycerol (4.42 g/kg, rabbit, National Library of Medicine: TOXNET) – page 8 line 195-197.

e. It will be of interest to define the context of these biological evaluation: is it for potential use in medical devices? In biomaterials conception? for food packaging ...etc.

Response 4e: Our aim with these materials is mainly for biomaterials and medical devices, although we find food packaging to be a very insightful application.

We have edited page 8 lines 189-192 to state “Although these polycarbonates are composed of relatively benign building blocks, evaluation of toxicity is warranted with the synthesis of new materials; especially given their potential use in the consumer goods (e.g., food packaging) and medical device areas (e.g., pressure sensitive adhesive).”

Minor Points

1- Page 3 Line 62 : “...including those derived from biological feed stocks “lessening” the dependency” Please replace the term “lessening” by lowering.

2- SI Page 1 line 14-15: “Reagents were purchased from Sigma Aldrich and used as received. Propylene Oxide was purchased from Sigma Aldrich”.

No need to mention Sigma Aldrich twice

3- SI In the ligand of Figure S7 please replace the term: “DSC trace” by “DSC thermogram”

4- SI Page 2 DSC paragraph: Authors state that “The measurements were performed from –70 °C to 125 °C at heating and cooling rates –70 °C to 125 °C and 2 lines further they state that “All samples... were tested from -60 to 100 °C). In figure S7 (DSC trace) the thermograph shown is from -50 to 80 °C. These values need to be homogenized. This paragraph needs to be properly reedited. There is multiple repetitions.

5- SI: In all NMR figures please add the term spectrum (NMR spectrum).

Response to Minor Points: Thank you for your careful read. We have made all the changes as suggested in the revised manuscript and SI.

Reviewer #2 (Remarks to the Author):

This manuscript by Grinstaff and coworkers describes an interesting and important study regarding the thermo-responsive sustainable polycarbonate adhesives, prepared from the terpolymerization of propylene oxide, glycidyl butyrate, and CO₂ using a bifunctional cobalt catalyst. Of the synthesized terpolymers, PPGBC-56 exhibited stronger adhesion than commercial scotch tape and comparable adhesion to duct tape. Notably, this adhesive sticks to a variety of chemically distinct materials, and its high viscosity at ambient temperature enables a thermo-responsive temperature trigger of adhesion to induce bonding and debonding. The work

was well done and good described. Therefore, I would like to recommend its publication in Nature Communication as it is or after minor revision.

Response: We thank the referee for their positive comments and recommendation for publication.

Minor points:

Comment 1. From the results in Table 1 at different monomer feed ratios of propylene oxide vs. glycidyl butyrate, glycidyl butyrate exhibited higher reactivity than propylene oxide during the their terpolymerization with CO₂ using the bifunctional cobalt catalyst. On the contrary, propylene oxide showed significantly greater reactivity than glycidyl butyrate in the copolymerization with CO₂. It means that the presence of propylene oxide enhances the reactivity of glycidyl butyrate, while glycidyl butyrate significantly inhibits the reactivity of propylene oxide. The authors should give a possible explanation.

Response 1: This is a very interesting and important point. The explanation for this result is the electronics of the pendant ester. The ester functionality of the GB monomer provides more electron density compared to the methyl group of the PO monomer, making the GB epoxide oxygen molecule more nucleophilic.

The reactivity of the catalyst depends upon the ability of the epoxide to coordinate with the catalyst active site as well as its ability to leave, making room for more chains to coordinate and propagate. In neat solutions of oversaturated monomers, the rate-limiting step becomes the detachment of the monomer from the catalyst rather than the coordination. PO polymerizes quicker than GB as observed by the kinetics of the polymerization. In these cases, the more nucleophilic GB takes longer to detach from the catalyst active site, as the additional electron density enhances the bonding to the cobalt metal center, leading to the observed slower polymerization rates. In the co-monomer mixture, the coordinating parameter comes into play. The more nucleophilic GB coordinates faster to the catalyst active site than PO. This leads to a preference of more GB units inserted in the polymer chain initially as observed by the Fineman-Ross analysis. The lowering of the catalyst reactivity upon the addition of GB to PO is the other aspect of this phenomenon.

In the revised manuscript, we have added to page 4 line 102-104 to state “The monomeric reactivity ratios for GB ($r_{GB} = k_{11}/k_{12}$) and PO ($r_{PO} = k_{22}/k_{21}$) are 1.32 and 0.26, respectively (**Figure 2**), indicating consecutive incorporation of two GB units is more favored during the terpolymerization. This preference is due to the electron donating effects of the pendant ester of GB, increasing epoxide nucleophilicity over PO, and enabling faster coordination to the catalyst active site.’

Comment 2. For a comparison purpose, it is suggested to synthesize PPC-co-PGBC block terpolymer for testing its adhesive property.

Response 2: We thank the referee very much for this insightful question. The block-co-polymer will likely provide additional insight into the adhesion mechanism. Due to the nature of this polymerization, the only way to prepare such a polymer architecture is to first synthesize a PGBC copolymer (CO₂ and GB epoxide). Purify this copolymer, then utilize it as a macro-initiator in the next polymerization step consisting of PO and CO₂ to form the block copolymers. However, this type of system suffers from some limitations.

- 1.) The PGBC polymer will possess both DNP and hydroxyl initiating groups, and hydroxyls as end groups. As separation would be extremely time consuming if even possible, this means both AB and ABA copolymers will form in the same mixture, as the hydroxyl is

the chemical moiety for propagation. This introduces an undesirable variable which will hinder proper analysis for comparison.

- 2.) Since the backbone of these polymers is an electrophilic carbonyl, anionic initiators in the second polymerization can attack the backbone at any segment and start a new propagation site. This leads to mixtures of random block segments in the polymer such that uniform polymers are not possible.

Thus, we will investigate other polymerization methods or monomers to access these block copolymers as they would be interesting materials to study.

Comment 3. Did the authors attempt other epoxides for the substitution for propylene oxide in the terpolymerization?

Response 3: We have substituted propylene oxide for glycidyl acetate and prepared a terpolymer of CO₂, glycidyl acetate, and glycidyl butyrate. Terpolymers are readily synthesized and the reactivity ratios for GA and GB are approximately $r_{GA}=0.66$ and $r_{GB}=0.34$. This work is ongoing in the laboratory and will be submitted for publication at a later date.

Comment 4. The used bifunctional cobalt catalyst in this study was first reported in 2009 (J. Am. Chem. Soc. 2009, 131, 11509–11518). Therefore, the supplemental citation of the literature is suggested.

Response 4: We thank the referee for pointing out this oversight. We have added this reference in the revised manuscript on page 2 line 59. We also added the reference and the specific catalyst used on page 3 line 80 of the text: “The catalyst, [S,S]-[SalcyCo^{III}DNP]/DNP,³⁸ polymerized PO with high turn-over frequency.....”

Reviewer #3 (Remarks to the Author):

The manuscript of Prof. Grinstaff et al. reports on the library of new, environmentally friendly, sustainable, strong, and responsive adhesives composed of carbonate terpolymers. The reliable bonding between soft tissues is challenging because of the wet and complex environment and geometries. Also available tissue adhesives do not meet basic demands strong adhesion and cohesion; controlled and precise delivery; biocompatibility and biodegradability. Therefore, the topic of the work on development of poly(propylene-co-glycidyl butyrate carbonate)s as sustainable polymer adhesives with an eco-design and performance in both dry and wet environments is very timely and crucially important. Important is that this polymer is non-toxic and non-immunogenic in vitro. Also the aspects of adherence to a variety of substrates such as metal, glass, wood, and Teflon® surfaces are imperative. Furthermore, reversibility and T-responsiveness of bonding between 21°C and 37°C-are addressed. The authors also have shown that adhesion increased at 37°C and is lost at 50°C. This effect should be more clearly explained in the discussion and conclusion part before publishing.

Response: We thank him/her for their high regard of our work, highlighting the thermo-responsive adhesive properties and adhesion to diverse substrates, and noting the importance and timeliness of the work.

To further explain the adhesive effects at 37 and 50 degrees Celsius, we have expanded the discussion and amended page 5-6 lines 136 to 149 to state:

“Specifically, as shown in Fig. 3c, the tack strength of PPGBC-56 with 1 N of applied axial pressure and 5 second dwell time at 20 °C is 9.0 ± 1.8 N in a dry environment, and the PSA exhibits adhesive failure with debonding occurring at the polymer-metal interface. Raising the temperature to 37 °C significantly increases the S_{tack} to 30.4 ± 5.2 N. At the higher temperature,

the polymer flows, spreads across, and wets the metal increasing the surface area of interaction and adherence to the metal rod surface. The interfacial adhesion between the two surfaces increases such that the PSA now fails cohesively (weaker interchain Van der Waals forces), and de-bonding occurs between the polymer strands. A further increase in temperature to 50 °C reduces the S_{tack} to 9.6 ± 1.6 N and by 100 °C, the tack strength is significantly less and 3.3 ± 0.2 N. As the temperature rises, interchain Van der Waals forces further weaken due to increased volume and mobility between polymer strands, and the energy required to de-bond the materials decreases. The same trend is observed with the PPGBC-56 in an aqueous environment with diminished tack strength overall.”

REVIEWERS' COMMENTS:

Reviewer #1 (Remarks to the Author):

I have read the answers, arguments and manuscript modifications that were presented by the authors. I consider that they properly treated the issues that I have raised in my initial review. Their scientific arguments about mechanism reaction and the molecular weight distribution are scientifically sound.

I recommend the publication of the manuscript in its revised version

Reviewer #2 (Remarks to the Author):

The authors have carefully revised the manuscript according to the referees' suggestions with some supplemental experiments, and given the detailed responses to the comments by points to points. Therefore, I suggest its publication in Nature Communications as the revision version is.

Reviewer #3 (Remarks to the Author):

As suggested, the authors expanded the discussion about the adhesive effects at 37 and 50 degrees Celsius. I have no other comments on the manuscript and suggest it for publishing in the present form.